# Loneliness trajectories over three decades are associated with conspiracist worldviews in midlife

Kinga Bierwiaczonek [1] ✉, Sam Fluit [1], Tilmann von Soest [1,2], Matthew J. Hornsey[3] & Jonas R. Kunst [1]

In the age of misinformation, conspiracy theories can have far-reaching consequences for individuals and society. Social and emotional experiences throughout the life course, such as loneliness, may be associated with a tendency to hold conspiracist worldviews. Here, we present results from a population-based sample of Norwegians followed for almost three decades, from adolescence into midlife ($N = 2215$). We examine participants' life trajectories of loneliness using latent growth curve modeling. We show that people reporting high levels of loneliness in adolescence, and those who experience increasing loneliness over the life course, are more likely to endorse conspiracy worldviews in midlife.

While conspiracy theories are not new[1,2], recent events have shown how dangerous and polarizing they can be in a globalized, mediatized world. Conspiracy theories undermined global efforts to contain the COVID-19 virus during the pandemic[3,4] and were used in the lead-up to the January 6, 2021, raid on the Capitol[1]. They lie at the core of political and social polarization[5,6], fueling vaccine skepticism[7], climate change skepticism[8,9], and anti-science movements such as the flat earthers[10,11]. In the age of misinformation[12], understanding what makes people endorse conspiracy theories is crucial. However, research on the psychology of conspiracy beliefs is rather recent, with more than half of the studies dating from 2020 or later[1]. Critically, very little longitudinal research on the antecedents of conspiracy beliefs is available to date. Existing studies capture only short periods of time[13,14], complicating the identification of early antecedents. Developmental perspectives examining how people's life trajectories are associated with conspiracy mindsets are therefore missing due to the lack of suitable data[1]. Here, we address this gap by investigating the link between conspiracy beliefs and loneliness trajectories over the course of three decades.

While several motives may be implicated in the development of conspiracist worldviews[15–18], both theory and research suggest that frustration of social needs and the resulting feelings of loneliness may be particularly important. A recent meta-analysis found that the factor showing the strongest cross-sectional association with conspiracy beliefs ($r = 0.37$) is social alienation, of which loneliness can be seen as a

facet[16]. Loneliness was also positively correlated with a conspiracy mindset ($r = 0.19$) in a representative German sample[19]. So far, however, research on the link between loneliness and conspiracy beliefs[16], and in particular longitudinal research that could clarify how this link plays out over time, is scarce[1]. Whether loneliness experienced in critical periods (i.e., during adolescence) and over prolonged periods is associated with a conspiracist worldview later in life has not been examined.

There are, however, several theoretical reasons for such an association. First, conspiracy beliefs may help make sense of one's loneliness[20] in a way that protects the ego, following general processes of motivated reasoning[21,22]. Sense-making and ego defense seem to be among the main psychological functions of conspiracy beliefs[23] and could be particularly relevant for lonely people who generally seem hypervigilant to social threats and may use blame to deal with their own negative emotions[24]. Conspiracy beliefs may preserve a positive self-image by shifting the blame for one's loneliness to malicious others (e.g., I am not a failure but a victim of a conspiracy)[17]. These beliefs may even enhance people's self-image by explaining their loneliness with their uniqueness (e.g., I am alone because I understand things others do not understand)[17,25]. Second, lonely people may lack the social feedback that could correct their developing conspiracist views, and once these views are formed, such people may purposefully seek reinforcing feedback from other like-minded conspiracy believers[26,27].

[1]Department of Psychology, University of Oslo, Blindern, Oslo, Norway. [2]Norwegian Social Research (NOVA), Oslo Metropolitan University, Oslo, Norway. [3]Business School, University of Queensland, Brisbane, QLD, Australia. ✉e-mail: k.m.bierwiaczonek@psykologi.uio.no

Lastly, loneliness may motivate people to adopt conspiracy beliefs in an attempt to gain community and a sense of social identity[1,28]. Several theoretical models describe loneliness as a motivational force across development[29–31]. Some people who see themselves as lonely may experience a motivation to reconnect[29,31], and seeking conspiracist communities might offer this opportunity. Online conspiracist groups in particular are easy to join, highly reinforcing and engaging, which may make them an accessible and suitable source of social nourishment and identity for socially isolated individuals[1,27,32]. Indeed, individuals high in conspiracy beliefs are those who feel most socially isolated after unplugging from the internet[33]. However, it should be noted that other lines of research suggest that loneliness is associated with social withdrawal rather than a motivation to reconnect[34,35], which would make this a less plausible mechanism underpinning the link between loneliness and conspiracy beliefs than the previously described mechanisms of ego protection and lack of corrective feedback.

In this work, we show that people's early experiences of loneliness and the increase of it throughout adulthood are positively associated with conspiracist worldviews in midlife. To do so, we use data collected from a population-based sample of 2215 Norwegians followed over 28 years.

## Results

Participants were junior and senior high school students in grades 7-12 ($M_{age} = 15.05$, $SD_{age} = 1.98$) at the first timepoint (in 1992), and in their early to mid-forties at the last timepoint (in 2020; $M_{age} = 43.22$, $SD_{age} = 2.00$). They reported their levels of loneliness at five timepoints between 1992 and 2020 on a Norwegian short version of the UCLA Loneliness Scale[36,37]. At the last timepoint, in their mid-forties, participants also reported to what extent they endorsed a conspiracist worldview assessed by the Conspiracy Mentality Questionnaire[38]. Both measures showed satisfactory reliability (.76 ≤ $α_{Loneliness}$ ≤ .80, $α_{CMQ} = .83$), and the loneliness measure was invariant over the five measurements, with the strong invariance model showing a close fit with the data, $\chi^2(70) = 29.36$, $p < .001$; CFI = .98; SRMR = .031; RMSEA = .038, $p_{close} = 1.000$, 90% CI$_{RMSEA}$ = [.033, .042]. Thus, we estimated a second-order latent growth curve model based on the strong invariance model (see Supplementary Table 3 for a technical description and detailed results).

Overall, consistent with previous research using the same scale[39] and cohort studies in Norway[40], participants' loneliness showed an increasing trajectory between 1992 and 2020 (see Supplementary Information for details). This increase was linear in shape: loneliness grew steadily from adolescence until mid-adulthood. We then tested if the initial level of loneliness (i.e., the intercept) and its trajectory over 28 years (i.e., the slope) were associated with participants' endorsement of a conspiracist worldview in midlife in a conditional growth model (Fig. 1, Table 1). Indeed, the lonelier participants were as adolescents in 1992, the greater their conspiracist worldview as adults in 2020, $z_{intercept} = 3.39$, $p = 0.001$, $β_{intercept} = 0.11$, 95% CI = [0.05, 0.18]. Moreover, the more loneliness increased over participants' life course, the more likely they were to report a conspiracist worldview in 2020, $z_{slope} = 4.00$, $p < 0.001$, $β_{slope} = 0.17$, 95% CI = [0.08, 0.25]. These results proved robust after controlling for age, sex, parental education, and political orientation measured in 1994, $z_{intercept} = 4.14$, $p < 0.001$, $β_{intercept} = .14$, 95% CI = [0.08, 0.19], $z_{slope} = 3.61$, $p < 0.001$, $β_{slope} = 0.15$, 95% CI = [0.08, 0.22] (Supplementary Table 6).

Separate lines of research have linked conspiracy beliefs[1,27] and loneliness[41] to psychopathology. To rule out that the associations with loneliness were artifacts of underlying psychopathology, we included symptoms of depression and anxiety measured by a short version of the Hopkins Symptom Checklist[42] (0.83 ≤ $α_{SCL}$ ≤ 0.91) as time-varying covariates in the model. Specifically, when estimating loneliness

growth curves, we regressed loneliness within each timepoint on depression and anxiety scores[43]; $\chi^2(324) = 2,134.06$, $p < 0.001$; CFI = 0.907, SRMR = 0.038; RMSEA = 0.050, $p_{close} = .424$, 95% CI$_{RMSEA}$ = [.048, .052]. In this model, the intercept represents the baseline, and the slope represents the growth in loneliness that is not attributable to temporal changes in symptoms of depression and anxiety. Even after controlling for these symptoms, the initial level, $z_{intercept} = 3.34$, $p = 0.001$, $β_{intercept} = .11$, 95% CI = [0.06, 0.16], and the trajectory, $z_{slope} = 2.42$, $p = 0.015$, $β_{slope} = 0.11$, 95% CI = [0.04, 0.19] (Supplementary Table 6) of loneliness remained positively related to conspiracy worldviews.

## Discussion

Our 28-year study shows that conspiracist worldviews held in midlife are associated with experiences of loneliness across adolescence and adulthood. Conspiracist worldviews were particularly appealing to participants who were relatively lonely as adolescents and experienced increasing loneliness through their lives. One possible explanation for this pattern, albeit tentative and requiring further research, is that the contrasting of one's own increasing loneliness relative to peers might be potent in fostering feelings of social isolation[44–46], motivating our participants to turn to conspiracy theorizing to protect their ego, or to seek social connection among like-minded conspiracist groups[1,17,20,28].

As in any observational study, we cannot exclude the possibility that our results are confounded by third variables, despite our efforts to control for age, sex, parental education, political orientation, and depression and anxiety. Factors such as personality traits, paranoid tendencies, and lower cognitive abilities, or experiences such as economic deprivation and selective media exposure might predispose individuals to both loneliness[47] and conspiracist worldviews[1,48–50]. Including all relevant variables in one observational study is infeasible, and only rigorous experimental research can further eliminate possible spuriousness. So far, experimental results show that manipulating ostracism increases participants' conspiracy beliefs, which aligns with our findings[51]. Yet, it is essential to highlight that loneliness and ostracism, albeit related, are distinct concepts that may allude to the unfulfillment of different social needs. Whereas ostracism refers to the interpersonal or intergroup process of deliberate exclusion, loneliness captures a socio-affective state that can arise from ostracism. However, loneliness can have numerous other causes[52–54], and not everyone who experiences ostracism or alienation necessarily feels lonely[55]. Therefore, future studies would benefit from exploring the nuanced impacts of ostracism, social alienation, loneliness, and their specific repercussions.

The associations observed in this study might also depend on contextual factors. Here, it is important to note that our data were collected in Norway—a technologically advanced society with high levels of institutional trust[56]—and future research is needed to test the generalizability of our findings in other contexts[54]. Norway can generally be described as a highly functioning welfare society with relatively low levels of loneliness[57]. On one hand, these low levels may mask, to some extent, the true size of associations between loneliness and conspiracist worldviews (i.e., due to floor effects). Thus, findings may be even more pronounced in societies where loneliness is more prevalent. On the other hand, given that loneliness seems generally uncommon in Norway, those who experience it might perceive themselves as outliers, prompting them to adopt ego defenses by embracing conspiracy theories. This may not be the case in societies where experiencing loneliness is more normative. Investigating the relationship between loneliness and conspiracist worldviews at the country level could bring further insights into the role of context, including cultural values (e.g., individualism-collectivism[1,54,58], and other nation-level variables such as general political climate, corruption, autocracy, or economic dysfunction[1,49].

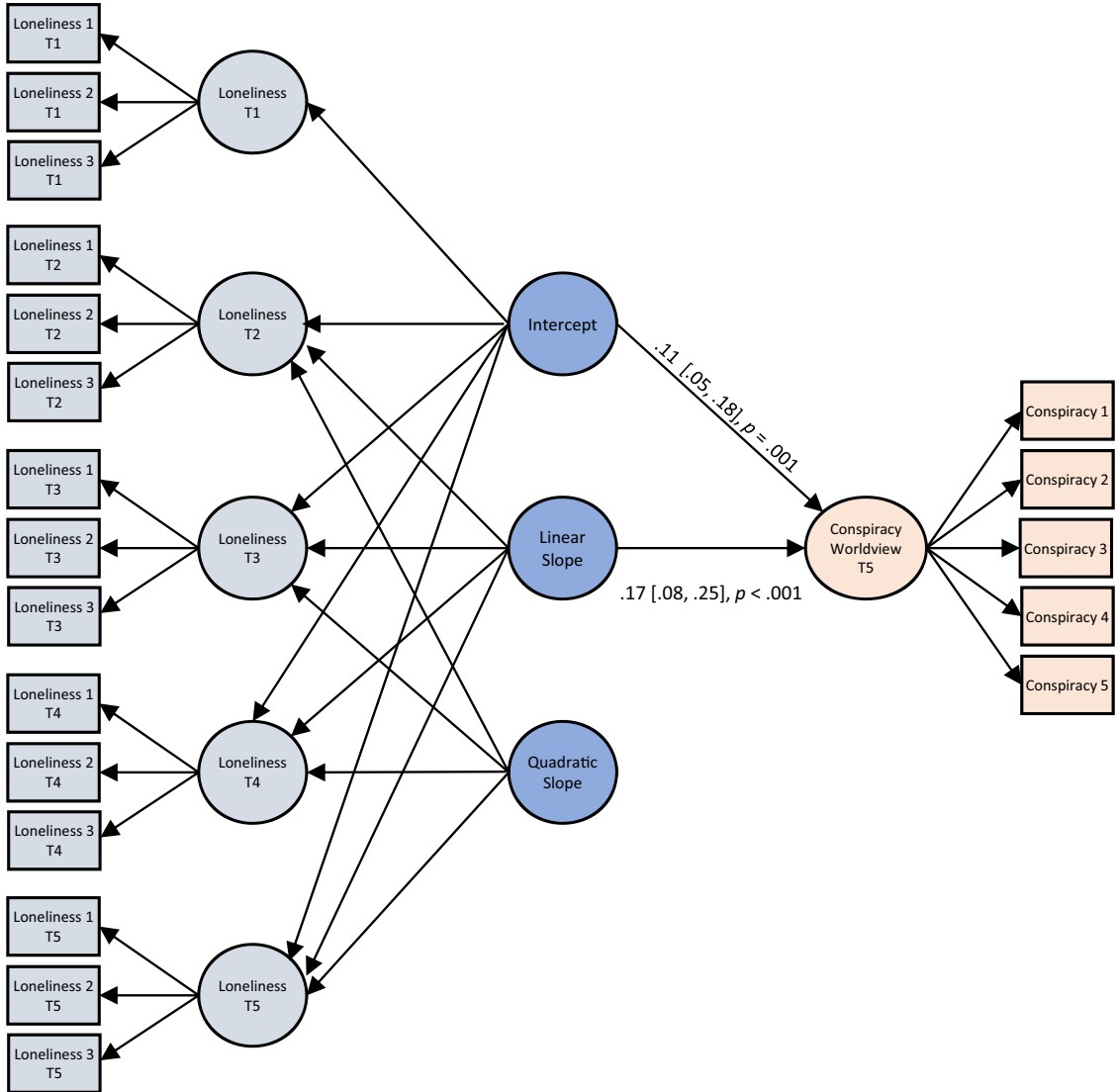

**Fig. 1 | Second order quadratic latent growth curve model showing that the initial level (intercept) of loneliness, as well as the trajectory (linear slope) of loneliness over five waves are associated with higher endorsement of a conspiracy worldview in midlife.** Standardized path coefficients, 95% confidence intervals and exact p-values are presented. Loneliness 1–3 and Conspiracy 1–5 denote the observed indicators (items) used to measure, respectively, loneliness and conspiracy worldview. T1–T5 denote five timepoints at which loneliness was measured. Measurement models of loneliness are displayed in light blue, second order latent constructs of the growth model of loneliness are displayed in dark blue, and the measurement model of conspiracy beliefs in pink. All correlations between the intercept and slopes, as well as correlations of residuals of equivalent items measuring loneliness at different timepoints, were also estimated but are not visualized for model readability. All p-values are two-tailed and based on the z statistic. Since the analysis is a latent growth curve model with one outcome, adjustments for multiple comparisons are not applicable. See Table 1 for detailed model estimates.

Research might also explore how effects differ when examining beliefs in specific conspiracy theories, oftentimes deeply rooted in culture and sociopolitical divides, as opposed to the broad conspiracist worldview our study focused on. Our broad approach avoids catering exclusively to specific cultural contexts or groups, thus enhancing the generalizability of our findings. However, it is possible that loneliness has stronger associations with some conspiracy beliefs than with others. For instance, conspiracy theories rooted in a nation's shared historical trauma are likely more commonly endorsed by its members[50,59] and thus, might be more appealing to those seeking social connection.

Methodologically, our study has one key limitation: as conspiracist worldviews represent a relatively recent construct in psychological research, controlling for participants' initial levels thereof in 1992 was impossible. Even so, we believe that our finding of a

significant association between loneliness in adolescence and conspiracy mentality in midlife goes beyond earlier cross-sectional findings[16,19]. Showing that the estimated levels of loneliness in adolescence are associated with conspiracy mentality, almost three decades later and accounting for later developments of loneliness, suggests that loneliness is related to conspiracy mentality over substantially long timeframes. Combined with the significant association of the slope of loneliness to conspiracy mentality in midlife, this finding may suggest that loneliness and conspiracy mentality are systematically related to each other from adolescence to midlife in ways consistent with the theoretical notion that conspiracist worldviews reflect sense making and ego defenses adopted in response to loneliness[20,21,23]. The unavailability of earlier measurements of conspiracy worldviews is a major limitation, nevertheless. Namely, this study design prevents us from testing a reverse causal direction: that adopting a conspiracist

**Table 1 | Results of second-order latent growth curve analyses**

| Model Estimates | Variable(s) | Estimate | SE | z | p | 95% CI |
|---|---|---|---|---|---|---|
| Means | | | | | | |
| | Linear Slope | 0.059 | 0.021 | 2.782 | 0.005 | [0.017, 0.100] |
| | Quadratic Slope | −0.008 | 0.007 | −1.188 | 0.235 | [−0.021, 005] |
| Variances | | | | | | |
| | Intercept | 0.300 | 0.017 | 17.750 | <0.001 | [0.267, 0.333] |
| | Linear Slope | 0.297 | 0.038 | 7.801 | <0.001 | [0.222, 0.371] |
| | Quadratic Slope | 0.018 | 0.005 | 3.797 | <0.001 | [0.009, 0.028] |
| Correlations | | | | | | |
| | Intercept with Linear Slope | −0.574 | 0.034 | −16.904 | <0.001 | [−0.641, −0.508] |
| | Intercept with Quadratic Slope | 0.543 | 0.067 | 8.131 | <0.001 | [0.412, 0.674] |
| | Linear Slope with Quadratic Slope | −0.980 | 0.047 | −20.978 | <0.001 | [−1.071, −0.888] |
| Regressions | | | | | | |
| | Intercept → Conspiracy Mentality T5 | 0.113 | 0.033 | 3.392 | 0.001 | [0.048, 0.179] |
| | Linear Slope → Conspiracy Mentality T5 | 0.165 | 0.041 | 4.002 | <0.001 | [0.084, 0.246] |
| Model Fit | | | | | | |
| | $X^2$ (150) = 827.911, $p < 0.001$ | | | | | |
| | CFI = 0.960 | | | | | |
| | TLI = 0.949 | | | | | |
| | RMSEA = 0.045, 90% CI$_{RMSEA}$ = [0.042, 048], $p_{close}$ = 0.996 | | | | | |
| | SRMR = 0.036 | | | | | |

Intercept – estimated initial level of loneliness; linear slope – linear change of loneliness (here, increase, as indicated by positive values); quadratic slope – the acceleration or deceleration of the change expressed by linear slope (i.e., the extent to which the decrease of loneliness slows down or accelerates over time); $\chi^2$ – chi-square, $df$ – degrees of freedom; CFI – comparative fit index; TLI – Tucker-Lewis index; SRMR – standardized root mean squared residual; RMSEA – root mean square error of approximation; 90% CI$_{RMSEA}$ – 90% confidence interval around RMSEA; $p_{close}$ – $p$-value of close fit testing the null hypothesis that RMSEA <.05 (i.e., that the model is close-fitting). For means and variances, unstandardized estimates are presented, whereas for correlations and regressions, standardized estimates are presented. All $p$-values of model estimates are two-tailed and based on the $z$ statistic. Since the reported analysis is a latent growth curve model with one outcome, adjustments for multiple comparisons are not applicable. The reported model does not include covariates (for detailed results of analyses with covariates, see Supplementary Tables 5, 6).

worldview might further exacerbate loneliness. Indeed, people who express conspiracy theories early in life might be excluded from social groups[1] which could lead to feelings of loneliness.

This considered, future research would be well advised to attempt a replication of our results in other contexts, and with experimental designs allowing for causal inferences. If successful, such replications would suggest that interventions targeting loneliness could be useful to reduce conspiracy beliefs and their societal repercussions. So far, the results of psychological interventions focusing primarily on cognitive processes (e.g., pre-bunking, debunking, cognitive inoculation) have been insufficient on their own to fully counter conspiracist worldviews[1,12,60]. There might, however, exist an alternative, complementary pathway to prevent conspiracy beliefs, one that leads via socio-affective processes. On one hand, previous research showed that the link between experimentally manipulated social exclusion (i.e., ostracism) and conspiracist thinking can be mitigated[51], and the same may be the case for loneliness. On the other hand, targeting loneliness and fostering social connection is known to be effective: it helps prevent other adverse outcomes, including somatic and mental health problems or even mortality risks, for a myriad of different social groups[61–64]. Therefore, instead of concentrating solely on cognitive factors, research may test experimentally whether reducing people's loneliness is a way to counter the onset of conspiracist worldviews and their societal repercussions.

## Methods
This study complies with all relevant ethical regulations for research with human subjects and obtained ethical approval from the Norwegian Regional Committees for Medical Research Ethics (reference no.: 25462; project name: Young in Norway). The study is based on survey data from the Young in Norway Study[65,66] collected at five timepoints: in 1992 (T1), 1994 (T2), 1999 (T3), 2005 (T4), and 2020 (T5). All items of

the multi-item measures used here are reported in Supplementary Table 1.

### Participants and procedure
At T1 (1992), a national sample of Norwegian junior and senior high school students, from 67 schools in grades 7–12 (age 12–20), was selected from stratified areas. Since the study did not include any experimental manipulations, no further randomization was applied. Each grade was equally represented, and cluster-sampling was applied with the school as the unit. See[67] for more information about the sampling procedures.

Data[68] were collected in the participating schools and the initial sample consisted of 11,985 participants, equally distributed according to gender and age. No statistical method was used to predetermine sample size. Written informed consent was obtained from participants or their parents whenever the participants were below the age of 15 at T1. Participants were followed up with questionnaires at school at T2 (1994), and a subset was then approached by postal means (using pen and pencil questionnaires) and digital means (using the Nettskjema online data collection tool) at the remaining time points: T3 (1999; $N = 2924$), T4 (2005; $N = 2890$), and T5 (2020; $N = 2215$). They were asked to renew their informed consent at T2 and at T4 in line with ethical stipulations.

Because the outcome measure of interest (i.e., conspiracy mentality) was included at T5, only data from participants who had completed this wave were used in the current study. Otherwise, no data were excluded from the analyses. In this sample, 57.4% of participants were women, and 42.6% were men. Most participants (93.6%) were ethnic Norwegians; 6.4% of participants had some immigrant background (i.e., were born abroad or had at least one parent who was born abroad). Less than half of participants (43.3%) had at least one parent who attended college or university.

**Table 2 | Estimated trajectories of loneliness by level of conspiracy worldview in midlife**

| | Intercept of Loneliness in 1992 | | | | Linear Slope of Loneliness | | | | Quadratic Slope of Loneliness | | | |
|---|---|---|---|---|---|---|---|---|---|---|---|---|
| | Estimate | z | 95% CI | p | Estimate | z | 95% CI | p | Estimate | z | 95% CI | p |
| High Conspiracy Worldview in 2020 | 0.231 | 3.348 | [0.088, 0.375] | 0.001 | 0.397 | 4.223 | [0.185, 0.609] | <0.001 | −0.019 | −2.714 | [−0.033, −0.006] | 0.003 |
| Moderate Conspiracy Worldview in 2020 | 0.004 | 4.000 | [0.002, 0.007] | 0.001 | 0.065 | 3.095 | [0.022, 0.108] | 0.002 | −0.008 | −1.143 | [−0.021, 0.005] | 0.222 |
| Low Conspiracy Worldview in 2020 | −0.223 | −3.379 | [−0.361, −0.085] | 0.001 | −0.267 | −2.967 | [−0.470, −0.065] | 0.003 | 0.004 | 0.571 | [−0.010, 0.017] | 0.592 |

Among participants who reported high levels of conspiracy mentality in mid-adulthood, estimated loneliness levels in adolescence were the highest (i.e., highest positive intercept value) and loneliness increased over time (i.e., positive linear slope). The increase in this group of participants was more rapid in adolescence and early adulthood, and decelerated later in life (i.e., negative quadratic slope). Among participants who reported moderate levels of conspiracy mentality in mid-adulthood, estimated loneliness levels in adolescence were close to the mean (i.e., intercept close to 0) and loneliness increased linearly across time (i.e., positive linear slope and non-significant quadratic slope). Among participants who reported low levels of conspiracy mentality in mid-adulthood, estimated loneliness levels in adolescence were the lowest (i.e., negative intercept), and loneliness decreased linearly across time (i.e., negative linear slope and non-significant quadratic slope). Thus, the greater increase in loneliness participants experienced over time, the more participants endorsed conspiracy worldviews in 2020. All p-values are two-tailed and based on the z statistic. Since the reported analysis is a latent growth curve model with one outcome, adjustments for multiple comparisons are not applicable.

## Analyses

**Descriptives and correlations.** Descriptive statistics and correlations between study variables are presented in Supplementary Table 2.

**Measurement invariance.** All analyses were conducted in Mplus v.8[69] and used two-tailed significance tests. Since latent growth curve models assume that modeled constructs are psychometrically equivalent across time, we first tested for longitudinal measurement invariance of the loneliness measure to assess whether this assumption was met. We report the results in Supplementary Table 3. We fitted a measurement model including all observed indicators of the latent construct of loneliness (i.e., the five items of the Norwegian short version of the UCLA loneliness scale[37]) at each timepoint (configural invariance model). We then constrained factor loadings to equality across timepoints (weak invariance model), and then loadings and intercepts of the items (strong invariance model). This analysis revealed that two out of the five initially used items (i.e., the reversed items, for which higher scores were thought to indicate less loneliness: "I feel in tune with the people around me", "I can find companionship when I want it") yielded low loadings on the latent loneliness construct ($0.29 \leq \beta \leq 0.54$) and were not invariant across time, resulting in poor fit of the strong invariance model, $\chi^2(251) = 2520.63$, $p < .001$, CFI = 0.880, TLI = 0.856, SRMR = 0.067, RMSEA = 0.064, $p_{close} < .001$, 90% $CI_{RMSEA} = [0.062, 0.066]$. Because latent growth modeling requires strong longitudinal invariance of the modeled construct[70], we removed these two reversed items, achieving excellent fit with the data for both weak invariance and strong invariance models. We therefore retained the strong invariance model based on the resulting three-item loneliness measure. Thus, in all remaining analyses, factor loadings and intercepts were constrained to equality across the five time points. Moreover, all models included correlations between residuals of the same items of loneliness measured at different timepoints.

**Latent growth curve analyses.** We fitted a series of second-order latent growth curve models, that is, models consisting of both the strong invariance measurement model and a latent growth curve model[71,72]. Supplementary Table 4 presents the detailed results of these analyses. To account for uneven time intervals between measurements, time for the slope components in these analyses was coded proportionally to the lag from the first measurement to each timepoint: T1 as 0, T2 as 0.2, T3 as 0.7, T4 as 1.3, and T5 as 2.8 (please note that decimals were used to avoid inflated variance values). Since the variances of all latent variables were set to 1 for model specification, loneliness values had a mean of zero and SD of 1. Full information likelihood estimation was used to handle missing values (≤10.2%).

First, we tested two univariate models including only loneliness at five time points to determine the shape of the trajectory: a linear model including the intercept and linear slope of loneliness (Model 1) and a quadratic model including the intercept, the linear slope, and the quadratic slope of loneliness that fitted the data best (Model 2). Then, to test whether the different trajectories of loneliness from adolescence into mid-adulthood are associated with conspiracy worldviews in 2020, we added conspiracy worldview at T5 as an outcome measure (again, as a measurement model with a latent construct of conspiracy worldview consisting of scale items as observed indicators) to the retained quadratic model (Model 3). We regressed conspiracy worldview on the intercept and linear slope of loneliness (but not on the quadratic slope, due to its high correlation with the linear slope and the resulting multicollinearity). Moreover, we tested the robustness of this model by adding time-invariant covariates (sex as recorded in national registries, age at T5 as recorded in national registries, political orientation at T2, parental education at T1; Model 4) and symptoms of depression and anxiety as time varying covariate (Model 5).

**Simple intercepts and slopes analysis.** We conducted simple intercepts and slopes analyses[73] to test how large the change in loneliness over three decades had been for participants who reported low, medium, and high levels of conspiracy worldview in midlife. Since the acceleration of the curve (quadratic slope) was not regressed on conspiracy worldview, we estimated its value at each level of conspiracy worldview based on the covariance of these two variables, using the formula:

$$q = \bar{q} + \frac{\text{cov}(q,x)}{\text{var}(q)}x \tag{1}$$

where q is the value of the quadratic slope at low, medium or high level of the outcome, $\bar{q}$ is the mean of the quadratic slope, cov(q,x) is the covariance between the quadratic slope and the outcome, var(q) is the variance of the quadratic slope, and x is the value of the outcome at the corresponding (low, $M - 1\,SD$; medium, $M$; or high $M + 1\,SD$) level of the outcome (here, conspiracy worldview in 2020). The results are presented in Table 2.

### Reporting summary
Further information on research design is available in the Nature Portfolio Reporting Summary linked to this article.

## Data availability
The primary data generated in this study have been deposited in the Open Science Framework repository under accession code https://osf.io/yjzqe (see Supplementary Information, Supplementary Note 1, for variable names). The raw data related to sociodemographic characteristics of the participants (age, gender, parental education) are protected and are not available due to Norwegian data privacy laws.

## Code availability

The analysis codes generated in this study have been deposited in the Open Science Framework repository under the accession code https://osf.io/yjzqe.

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

## Acknowledgements

This research was funded by the Research Council of Norway, grants #288083 and #301010 (T.v.S., S.F.). This research was supported by an EEA Grant 2014–2021 (Nr. 2019/35/J/HS6/03498) in the IdeaLab call operated by the National Science Centre (NCN).

## Author contributions

K.B., T.v.S., J.R.K., S.F., and M.J.H. conceived the original idea for this research. K.B., S.F., and T.v.S. developed the methodological approach. K.B. conducted the analyses. K.B. and J.R.K. visualized the results. T.v.S. acquired the funding for this study. K.B. drafted the original manuscript. K.B., J.R.K., S.F., T.v.S., and M.J.H. revised the manuscript and approved the final version.

## Competing interests

The authors declare no competing interests.
