## [Peer Review File · Nature Communications]

Loneliness trajectories over three decades are associated with conspiracist worldviews in midlifeREVIEWER COMMENTS

Reviewer #1 (Remarks to the Author):

The present paper reports a latent growth curve model of loneliness (measures at five waves spread over 28 years) predicting conspiracy mentality at t5. There is a lot to like about this paper, it is well written, highly accessible and the analyses seem to be conducted very carefully. The establishment of measurement variance across waves is laudable, the delineation of fitting models as well. Nevertheless, I have some remarks that might help the authors' in further refining their manuscript. The biggest issue to me is the fact that conspiracy mentality is only measured at t5. Although this is acknowledged by the authors, I had the feeling they do not really explicate how severe a limitation this is. At times, the authors seemed to interpret this association causally (e.g., "One major implication of our findings is that interventions targeting loneliness might reduce conspiracy beliefs and their societal repercussions"). This insinuation of support for a cause-effect relation is not warranted in light of the data. The data are very compatible with the notion that conspiracy mentality leads to loneliness (more and more over time, explaining the linear slope) or that a third variable causes both a trajectory of increasing loneliness and higher conspiracy mentality at t5 (e.g., economic deprivation leading to compensatory conspiracy beliefs and exclusion from social activities). As such, despite my admiration for the authors' diligent analyses, I am afraid that the inferences we can really draw from the results are not as strong as the writing seems to imply. The association with the intercept is at the level of cross-sectional associations that have been repeatedly shown in prior work: loneliness reports correlate positively with conspiracy mentality. The association with the slope seems more novel. Conspiracy mentality is correlated with an increase in loneliness over time – without any strong evidence for the causal order.

Minor issues:

Figure 2 presents the data in a way that seems more alignable with the reverse causal order: Loneliness as a function of conspiracy mentality. It would seem advisable to align the graphic representation with the theoretical rationale more tightly.

I wondered about the argument that quadratic slope and linear slope were too highly correlated to regress conspiracy mentality on both. I am by no means an expert in this kind of data modelling but it made me wonder whether it would not be more appropriate to then compare models with linear slope predictions and quadratic slope prediction to test which of the two fits the data better? I was also somewhat surprised to learn that the two slopes were extremely highly negatively correlated (Table 4: "Linear Slope with Quadratic Slope $-.973 .041 <.001 -1.053 .894$ " – by the way the upper bound of the CI misses a minus sign). Can that be explained or put into words what this implies?

Roland Imhoff (signed)

Reviewer #2 (Remarks to the Author):

Thank you very much for the opportunity to review the manuscript titled "Loneliness trajectories over three decades predict conspiracist worldviews in midlife". These findings are vital for the literature, and the methods employed are impressive and appear robust to inform the conclusions made.

I have some very minor recommendations, mostly with regards to clarifying concepts and recommendations on the back of your results:

1) Definitions

While I personally find the different conceptualisations of loneliness, alienation, social exclusion etc relatively intuitive, I'm not sure the distinctions and overlaps between these variables are really made clear to the reader. Most importantly, the meta-analytic evidence for alienation is perusasive, but leaves me wondering whether it is the same as loneliness. While alienation appears to refer more to society,

social exclusion appears to be more interpersonal. Some more consideration for the distinction between these variables would be really helpful I think, especially regarding similar evidence in the conspiracy beliefs literature (e.g., Hettich et al., 2022) that might support your findings.

2) Interventions

This may be a personal disagreement, but your mention of interventions on p. 6 did not quite appear to represent the literature in the same way that I see it. Especially with regards to pre-bunking, certain meta-analyses appear to have found effects sizes of around $d = 0.44$, which to me seems far from satisfactory. Furthermore, intervention evidence for social exclusion (ostracism) has been attempted by Poon et al. (2020), where they showed that self-affirmation neutralised the effect of ostracism on conspiracy beliefs. I believe these are things that might be worth incorporating as they may be less sparse or disheartening as you originally thought.

3) Cultural relevance

While I agree with your analysis of the Norwegian context, I think it would be nice to unpack this in a bit more detail. Since you mention that it may be more pronounced in other cultures, why might this be? It also might be worth speculating on the alternative account, as perhaps more chronic experiences of loneliness make this experience less potent in forming a conspiracy mindset due to the cultural normality of loneliness in certain cultures (or other reasons, just an example).

References

Hettich, N., Beutel, M. E., Ernst, M., Schliessler, C., Kampling, H., Kruse, J., & Braehler, E. (2022). Conspiracy endorsement and its associations with personality functioning, anxiety, loneliness, and sociodemographic characteristics during the COVID-19 pandemic in a representative sample of the German population. *PLOS ONE*, 17(1), e0263301. <https://doi.org/10.1371/journal.pone.0263301>

Poon, K., Chen, Z., & Wong, W. (2020). Beliefs in conspiracy theories following ostracism. *Personality and Social Psychology Bulletin*, 46(8), 1234-1246. <https://doi.org/10.1177/0146167219898944>

Reviewer #3 (Remarks to the Author):

Review of "Loneliness trajectories over three decades predict conspiracist worldviews in midlife"

Summary: This article presents a longitudinal study that examines how loneliness across adolescence and adulthood predicts conspiracy beliefs in midlife. The authors use data from a population-based sample of Norwegians followed over 28 years and find that higher levels of loneliness in adolescence and its persistence over the life course are positively associated with conspiracy beliefs in midlife. The authors suggest that loneliness may lead to conspiracy beliefs as a way of coping with social isolation, seeking social connection, and protecting one's self-image. They also propose that interventions targeting loneliness might reduce or prevent conspiracy beliefs and their negative consequences for individuals and society.

The article addresses an important and timely topic of conspiracy beliefs and their psychological antecedents. The authors use a unique data set that allows for examining the long-term effects of loneliness on conspiracy beliefs over almost three decades. They employ rigorous statistical methods to test for measurement invariance, latent growth curve models, and robustness checks.

Nonetheless, the article has some limitations that could be addressed or acknowledged by the authors. First, the article does not account for potential confounding or moderating factors that could affect the relationship between loneliness and conspiracy beliefs (e.g., personality traits, cognitive abilities, social support, media exposure, or cultural values). The authors could include some of these variables as covariates or moderators in their models or discuss their possible role in future research. Second, the

article does not provide a clear theoretical framework or mechanism for explaining how loneliness leads to conspiracy beliefs. The authors could elaborate on their hypotheses and draw on existing theories or models of loneliness, social cognition, or motivated reasoning to support their arguments. Third, the article does not address the generalizability or replicability of their findings to other populations or contexts. The authors could discuss the limitations of their sample (e.g., predominantly ethnic Norwegians, relatively low levels of loneliness) and the specificity of their context (e.g., Norway as a highly functioning welfare society) and suggest how their results might differ or apply to other groups or settings.

In sum, the article makes an interesting contribution to the literature on conspiracy beliefs and their psychological antecedents by using a longitudinal design and a large population-based sample. However, there are some methodological and conceptual limitations that weaken the study's validity and impact.

Reviewer #4 (Remarks to the Author):

Bierwaczzonek et al. "Loneliness trajectories over three decades predict conspiracist worldviews in midlife"

The manuscript summarizes an interesting study on the association between loneliness (measured longitudinally throughout adolescence) and conspiracist worldviews later in life. The authors find that high levels of loneliness, especially if they persist over time, predict conspiracist worldviews.

The study is interesting and novel and the methods used to address the research questions are advanced. Overall, I think this could be an interesting contribution to the existing literature on loneliness (and conspiracy beliefs). However, I have some concerns about the manuscript in its current version.

My biggest concern is related to the way how the variables were measured, particularly the fact that while loneliness was measured longitudinally over time, conspiracist worldview was not. In my opinion this is a major shortcoming of the present study because it hinders the authors to assess how longitudinal within-subject changes in one variable affect within-subject changes in the other, which would allow them to actually draw solid conclusions about predictions. As it is now, we don't know if individuals who showed conspiracist worldviews might have already held those at the beginning of data collection, which might have been the reason why they became lonely in the first place. Thus, there is no way to test if the causality is actually in the other direction. However, I don't think there is a way how the authors can address this alternative explanation given the data. I therefore suggest that the authors thoroughly rephrase the manuscript to tone down the implied causality of the results. At the moment the whole manuscript, and especially the discussion, heavily imply causality in the effects, which I don't think is warranted given the data.

Some parts of the data analysis were not clear to me, and I think it would be important to add more information there:

-lines 81-82 in the results section: the authors describe how they regressed loneliness on depression and anxiety scores thereby removing variance in loneliness explained by mental health problems. I don't fully understand how and why this regression removes variance coming from mental health problems and I would imagine other readers might find this hard to understand as well. It would be good to provide more explanation on this approach. Given that depression/anxiety are highly correlated with loneliness at each time point (as seen in table S2), this is not a trivial issue so it would be great to see more information on this part of the analysis. Related to that, I was wondering if there are interactions between loneliness and depression/anxiety that might explain developments of conspiracist world views. As in, there might be a difference in conspiracist worldviews between people

who are lonely and show high scores of depression/anxiety compared to those who are lonely but have low depression/anxiety. It would be interesting to explore such interactions.

-I don't understand why conspiracy worldview was regressed on the linear slope of loneliness but not the quadratic slope. The authors say that only the linear slope was used due to its high correlation with the quadratic slope to avoid multicollinearity, which I understand, but why did the authors not choose the quadratic slope instead of the linear slope in that case given that the quadratic slope model fitted the data best?

-it would be interesting to see data on how many people in the sample actually held conspiracy beliefs. I couldn't find this information in the manuscript or supplement (but might have missed it)

Introduction/Discussion:

-I was missing more background/ a theoretical framework on conspiracy worldviews in the introduction. For example, do we assume that people who hold conspiracy beliefs do it for similar reasons even if the specific conspiracies differ in terms of content?

-the authors state in the discussion (line 101) that individuals who are lonely experience a powerful motivation to reconnect. However, many studies within the loneliness literature show that lonely individuals often show lower social approach motivation. It would be nice to have a more nuanced discussion on why loneliness might affect conspiracy worldviews.

Reviewer #1:

***RI.1.** The present paper reports a latent growth curve model of loneliness (measures at five waves spread over 28 years) predicting conspiracy mentality at t5. There is a lot to like about this paper, it is well written, highly accessible and the analyses seem to be conducted very carefully. The establishment of measurement variance across waves is laudable, the delineation of fitting models as well. Nevertheless, I have some remarks that might help the authors' in further refining their manuscript.*

Response to RI.1. We thank the Reviewer for the positive evaluation and constructive feedback. Indeed, we feel that the manuscript benefitted a lot from addressing the Reviewer's suggestions.

***RI.2.** The biggest issue to me is the fact that conspiracy mentality is only measured at t5. Although this is acknowledged by the authors, I had the feeling they do not really explicate how severe a limitation this is. At times, the authors seemed to interpret this association causally (e.g., "One major implication of our findings is that interventions targeting loneliness might reduce conspiracy beliefs and their societal repercussions"). This insinuation of support for a cause-effect relation is not warranted in light of the data. The data are very compatible with the notion that conspiracy mentality leads to loneliness (more and more over time, explaining the linear slope) or that a third variable causes both a trajectory of increasing loneliness and higher conspiracy mentality at t5 (e.g., economic deprivation leading to compensatory conspiracy beliefs and exclusion from social activities). As such, despite my admiration for the authors' diligent analyses, I am afraid that the inferences we can really draw from the results are not as strong as the writing seems to imply.*

Response to RI.2. We thank the Reviewer for this comment. The fact that conspiracy mentality is only available at T5 is indeed a limitation of this study, and we will discuss it in detail in *Response to RI.3*. Here, we will focus on the issue of causality.

We agree that, as in any observational longitudinal study, our study only fulfills two conditions of causality - association and time precedence - but it cannot fulfill the third condition, non-spuriousness. Accordingly, we have taken various steps to address the Reviewer's valid comment. First, we have removed any language that can be interpreted as implying causality. Second, we now discuss more extensively the limitations of our analyses, including the role of confounding variables at different levels of analysis (p. 7):

As in any observational longitudinal study, we cannot exclude the possibility that our results are confounded by third variables. We attenuated these risks by controlling for several potential

confounds, namely age, sex, parental education, political orientation, depression and anxiety. Yet, other factors could play a role. For instance, individual differences such as personality traits, paranoid tendencies, lower cognitive abilities or experiences such as economic deprivation might predispose individuals to both loneliness (48) and conspiracist worldviews (1, 49-51). However, including all relevant variables in one observational study is infeasible, and only rigorous experimental research can further eliminate possible spuriousness.

Third, when discussing the potential role of interventions, we now remind readers of the need to replicate our findings within different contexts and with different designs and generally formulate our suggestions more cautiously (p. 9):

Although we encourage the replication of our findings in other contexts and with different designs to strengthen causal inferences, this study suggests that interventions targeting loneliness could be useful to reduce conspiracy beliefs and their societal repercussions.

RI.3. *The association with the intercept is at the level of cross-sectional associations that have been repeatedly shown in prior work: loneliness reports correlate positively with conspiracy mentality. The association with the slope seems more novel. Conspiracy mentality is correlated with an increase in loneliness over time – without any strong evidence for the causal order.*

Response to RI.3. We agree that our study does not provide strong evidence for a causal order, and we have revised the manuscript carefully to avoid causal claims. However, we believe that our finding of a significant association between the intercept of loneliness (representing the estimated level of loneliness in adolescence) and conspiracy mentality in midlife provides novel results that cannot be covered by cross-sectional findings. Showing that estimated levels of loneliness in adolescence are predictive of conspiracy mentality in midlife, almost three decades later, suggests that adolescent loneliness is related to conspiracy mentality over substantially long timeframes, even when accounting for several potential confounders in adolescence that are closely related to conspiracy mentality. This finding, combined with the significant association of the loneliness slope to conspiracy mentality in midlife, indicates that loneliness and conspiracy mentality are related to each other from adolescence to midlife in complex and systematic ways.

Please also note that the goal of the inclusion of autoregressive paths in longitudinal models is to control for the assumed stability of constructs (Adachi & Willoughby, 2015; Hertzog & Nesselrode, 1987). However, the assumption of stability is a strong assumption, as explained by Hertzog & Nesselrode (1987, p. 101):

This assumption appears to make sense for certain psychological phenomena, that is, those suspected to be enduring, such as stable attributes of individuals that have reached a determined end

state [...]. The assumption of inertial stability of individual differences modeled via autoregressive coefficients makes little sense for fluctuant attributes.

In our case, given that the study covers 28 years, assuming construct stability is unrealistic: it would require loneliness and conspiracy mindsets to have reached their “determined end state” already in adolescence to remain stable until mid-life, which would be at odds with developmental processes. Indeed, we showed that loneliness has low stability over this period (T1 only correlates with T5 at .28; Supplementary Materials, Table S2). With low stability of constructs, controlling for autoregressions is unlikely to meaningfully affect the results. Some statisticians would even argue that adding autoregressions in such a case is incorrect, as they claimed that autoregressions are overused for variables that cannot be expected to be stable (e.g., Hertzog & Nesselroade, 1987; Rogosa & Willet, 1985).

We would also like to note that latent growth curve models, used in this study, account for the associations between the outcome and the predictor within each time point by assuming that these associations are mediated by the growth factor (i.e., the mere passing of time between each measurement plays a role; Stoel et al., 2004). This assumption allows for developmental processes, which is ideal for our research question.

Therefore, we believe that our study has substantial strengths over cross-sectional studies. However, we also acknowledge that it would have been an advantage if conspiracy mentality had been measured in adolescence, and would have been tested as a confounder, in addition to the covariates that are currently included in the study. In the revised manuscript, we now emphasize this limitation on pp. 6 - 7:

Methodologically, our study has one key limitation: as conspiracist worldviews represent a relatively recent construct in psychological research, controlling for participants' initial levels in 1992 was impossible. Even so, we believe that our finding of a significant association between loneliness in adolescence and conspiracy mentality in midlife is a novel result that goes far beyond earlier cross-sectional findings (16, 34). Showing that the estimated levels of loneliness in adolescence are predictive of conspiracy mentality in midlife, almost three decades later and accounting for later developments of loneliness, suggests that adolescent loneliness is related to conspiracy mentality over substantially long timeframes. Notably, this finding held even when controlling for several potential confounders closely related to conspiracy mentality, such as political orientation (5) and anxiety and depression (1). Combined with the significant association of the slope of loneliness to conspiracy mentality in midlife, this finding indicates that loneliness and conspiracy mentality are systematically related to each other from adolescence to midlife in ways consistent with the theoretical notion that conspiracist worldviews reflect sense making and ego defenses adopted in response to loneliness (19, 20, 22). The unavailability of earlier measurements of conspiracy worldviews is a limitation, nevertheless. For instance, the study design prevents us from testing a plausible feedback loop: that adopting a conspiracist worldview might further exacerbate loneliness. Indeed, conspiracists might be excluded from many social

groups or ostracized (1), which would indicate that turning to conspiracy beliefs is not an adaptive strategy to cope with loneliness.

References:

- Adachi, P., & Willoughby, T. (2015). Interpreting effect sizes when controlling for stability effects in longitudinal autoregressive models: Implications for psychological science. *European Journal of Developmental Psychology, 12*(1), 116-128.
- Hertzog, C., & Nesselroade, J. R. (1987). Beyond autoregressive models: Some implications of the trait-state distinction for the structural modeling of developmental change. *Child Development, 93*-109.
- Rogosa, D., & Willett, J. B. (1985). Satisfying a simplex structure is simpler than it should be. *Journal of Educational Statistics, 10*(2), 99-107.
- Stoel, R. D., van den Wittenboer, G., & Hox, J. (2004). Including time-invariant covariates in the latent growth curve model. *Structural Equation Modeling, 11*(2), 155-167.

RI.4. Minor issues: *Figure 2 presents the data in a way that seems more alignable with the reverse causal order: Loneliness as a function of conspiracy mentality. It would seem advisable to align the graphic representation with the theoretical rationale more tightly.*

Response to RI.4. We thank the Reviewer for this suggestion. We agree that the way conditional trajectories were presented in previous Fig. 2 could be confusing to readers used to classic moderation graphs. Unfortunately, since the analysis presented here is rather rare, we failed to find alternative ways of visualizing it in the literature. Therefore, we have now removed Fig. 2 entirely and instead moved the table reporting the full results of simple intercepts and slopes analysis from the supplementary materials to the main manuscript (former Table S5, now Table 1). To ensure that the results are clear to readers, we added an extensive note explaining their interpretation. We believe that this way of reporting simple trajectories has an advantage over a figure as it is more detailed and less prone to misinterpretations.

Please note that the results of this analysis have slightly changed after improving model specification in line with most recent guidelines for second-order latent growth curves (Wickrama et al., 2022; namely, removing intercept constraints within each wave in the strong invariance model, and maintaining intercept constraints between waves). By consequence, the average loneliness trajectory, as well as the trajectory of participants who ended up with high level of conspiracy mentality in 2020, is now increasing. Importantly, however, the core results (i.e., the association between loneliness across the life course and conspiracy mentality) remained unchanged, speaking to its robustness.

References:

- Wickrama, K. A., Lee, T. K., O'Neal, C. W., & Lorenz, F. O. (2022). *Higher-order growth curves and mixture modeling with Mplus: A practical guide*. Routledge.

RI.5. *I wondered about the argument that quadratic slope and linear slope were too highly correlated to regress conspiracy mentality on both. I am by no means an expert in this kind of data modelling but it made me wonder whether it would not be more appropriate to then compare models with linear slope predictions and quadratic slope prediction to test which of the two fits the data better?*

Response to RI.5. We appreciate this comment. We indeed tested and compared models with linear and quadratic predictions, and we report the results in Table S4 (Model 1, 2). These results indicated a better fit for the quadratic model, $\Delta\chi^2(4) = 76.233, p < .001$. We would also like to clarify that, although the overall quadratic slope became non-significant after model re-specification ($p = .235$ in the core analysis), its inclusion was still required for model fit. The reason becomes clear when considering the simple slope analysis: Indeed, there was a significant deceleration in loneliness among those participants for whom loneliness was increasing (i.e., those with high levels of conspiracy beliefs in 2020), but not among other participants, which explains why adding it improved the model. We thus retained it.

RI.6. *I was also somewhat surprised to learn that the two slopes were extremely highly negatively correlated (Table 4: “Linear Slope with Quadratic Slope $-.973 .041 <.001 -1.053 .894$ ” – by the way the upper bound of the CI misses a minus sign). Can that be explained or put into words what this implies?*

Response to RI.6. We thank the Reviewer for drawing our attention to this point. As described in the methodological literature (see, e.g., Newsom, 2015, p. 223), the strong negative association between the linear slope and the quadratic slope stems directly from latent growth curve models’ specification. The linear slope is the 1st degree polynomial and indicates whether the trend in the data is decreasing or increasing; the quadratic slope is the 2nd degree polynomial that represents the acceleration/deceleration of this trend. Both pieces of information describe a single trajectory line that can be expressed as:

trajectory = intercept + linear slope + quadratic slope + error

It stems from the specification of the model that the linear and quadratic slope are highly correlated because they are, by design, based on transformations of the same variables. For example:

Linear slope = $\sim \mu_{T1} * 0 + \mu_{T2} * 1 + \mu_{T3} * 2 + \mu_{T4} * 3$

Quadratic slope = $\sim \mu_{T1} * 0^2 + \mu_{T2} * 1^2 + \mu_{T3} * 2^2 + \mu_{T4} * 3^2 = \mu_{T1} * 0 + \mu_{T2} * 1 + \mu_{T3} * 4 + \mu_{T4} * 9$

To conclude, the intercept and slope factors will be highly correlated because the quadratic time values are a multiplicative function of the linear time values (Newsom, 2015). Thus, correlations over .90 can be expected.

As to the sign of these correlations, it follows from the above specification that if the linear slope is increasing (i.e., means at the subsequent timepoints are getting larger, like in our case), its sign will be positive, whereas if it is descending its sign will be negative. The quadratic slope, as the 2nd order polynomial, will indicate whether this ascent/descent is accelerating (when its sign is the same as the linear slope) or decelerating (when its sign is opposite than the linear slope, as in the present paper). Thus, by design, when the trajectory is increasing, and this increase decelerates over time (as it does in our case, although this deceleration is now only significant for participants with high levels of conspiracy mentality, see Table 1), the correlation between the linear slope and the quadratic slope will be negative.

Lastly, we would like to thank the Reviewer for noticing the missing minus sign in Table S4. We have now corrected this mistake and checked all the tables to make sure that similar typos do not appear in other parts of the results.

Reference:

Newsom, J. T. (2015). *Longitudinal structural equation modeling. A comprehensive introduction*. Routledge.

Reviewer #2:

***R2.1.** Thank you very much for the opportunity to review the manuscript titled "Loneliness trajectories over three decades predict conspiracist worldviews in midlife". These findings are vital for the literature, and the methods employed are impressive and appear robust to inform the conclusions made.*

I have some very minor recommendations, mostly with regards to clarifying concepts and recommendations on the back of your results.

Response to R2.1. We very much appreciate the Reviewer's positive and encouraging evaluation, as well as their constructive feedback.

R2.2. 1) Definitions

While I personally find the different conceptualisations of loneliness, alienation, social exclusion etc relatively intuitive, I'm not sure the distinctions and overlaps between these variables are really made clear to the reader. Most importantly, the meta-analytic evidence for alienation is persuasive, but leaves me wondering whether it is the same as loneliness. While alienation appears to refer more to society, social exclusion appears to be more interpersonal. Some more consideration for the distinction between these variables would be really helpful I think, especially regarding similar evidence in the conspiracy beliefs literature (e.g., Hettich et al., 2022) that might support your findings.

Response to R2.2. We thank the Reviewer for this comment and for drawing our attention to this study, which we now have cited in the introduction. Indeed, in the previous version of the manuscript, we used several related but not entirely synonymous concepts (alienation, loneliness, ostracism) without much explanation, which might have been confusing to readers. As suggested, we now have acknowledged the difference between loneliness and alienation already in the introduction (p. 3):

Supporting these notions, a recent meta-analysis found that the factor showing the strongest cross-sectional association with conspiracy beliefs ($r = .37$) is social alienation, of which loneliness can be seen as a facet (16).

Crucially, we now discuss differences between loneliness and ostracism, highlighting interesting avenues for future research (p. 7):

So far, experimental results align with our findings, showing that manipulating ostracism increases participants' conspiracy beliefs (52). Yet, it is essential to highlight that loneliness and ostracism, albeit related, are distinct concepts that may allude to the unfulfillment of different social needs. Whereas ostracism refers to the interpersonal or intergroup process of deliberate exclusion, loneliness captures a socio-affective state that can arise from ostracism. However, loneliness can have numerous other causes (53-55), and not everyone who experiences alienation or ostracism necessarily feels lonelier (56). Therefore, future studies would benefit from exploring the nuanced impacts of social alienation, ostracism, and loneliness, and their specific repercussions.

Moreover, we have replaced “social isolation” with “loneliness” at all times to prevent confusion.

R2.3. 2) Interventions

This may be a personal disagreement, but your mention of interventions on p. 6 did not quite appear to represent the literature in the same way that I see it. Especially with regards to pre-

bunking, certain meta-analyses appear to have found effects sizes of around $d = 0.44$, which to me seems far from satisfactory. Furthermore, intervention evidence for social exclusion (ostracism) has been attempted by Poon et al. (2020), where they showed that self-affirmation neutralised the effect of ostracism on conspiracy beliefs. I believe these are things that might be worth incorporating as they may be less sparse or disheartening as you originally thought.

Response to R2.3. We thank the Reviewer for this comment, which encouraged us to refine our discussion of interventions. We agree that the outcomes of interventions utilizing cognitive processes, such as pre-bunking, to counter conspiracist worldviews are insufficient on their own. This underscores the necessity for alternative interventions. In the revised paragraph, we now discuss this issue in light of Poon et al. (2020) as suggested (p. 9):

Although we encourage the replication of our findings in other contexts and with different designs to strengthen causal inferences, this study suggests that interventions targeting loneliness could be useful to reduce conspiracy beliefs and their societal repercussions. So far, the results of psychological interventions focusing primarily on cognitive processes (e.g., pre-bunking, debunking, cognitive inoculation) have been insufficient on their own to fully counter conspiracist worldviews (1, 12, 61). Our findings point to an alternative pathway to prevent conspiracy beliefs, one that leads via socio-affective processes. On the one hand, previous research showed that the link between experimentally manipulated social exclusion (i.e., ostracism) and conspiratorial thinking can be mitigated (52), and the same may be the case for loneliness. On the other hand, targeting loneliness and fostering social connection is known to be effective: it helps prevent other adverse outcomes, including somatic and mental health problems or even mortality risks, for a myriad of different social groups (62-65). Therefore, instead of concentrating solely on cognitive factors, research should explore if interventions reducing people's loneliness throughout the lifespan may effectively counter or even prevent the onset of conspiracist worldviews and their societal repercussions. Since effective interventions against loneliness exist (64, 66-68), a possible antidote to conspiracist worldviews and their negative consequences might already be in our hands.

R2.4. 3) Cultural relevance

While I agree with your analysis of the Norwegian context, I think it would be nice to unpack this in a bit more detail. Since you mention that it may be more pronounced in other cultures, why might this be? It also might be worth speculating on the alternative account, as perhaps more chronic experiences of loneliness make this experience less potent in forming a conspiracy mindset due to the cultural normality of loneliness in certain cultures (or other reasons, just an example).

Response to R2.3. We appreciate this excellent comment. We agree with the Reviewer that the cultural context is important. We have now added a more extensive discussion of it and acknowledged the role of country-level and cultural variables, on p. 8:

The associations observed in this study might also depend on contextual factors. Here, it is important to note that our data were collected in Norway – a technologically advanced society with high levels of institutional trust (57) – and future research is needed to test the generalizability of our findings in other contexts (55). Norway can generally be described as a highly functioning welfare society with relatively low levels of loneliness (58). On the one hand, these low levels may mask, to some extent, the true size of associations between loneliness and conspiracist worldviews (i.e., due to floor effects). Thus, findings may be even more pronounced in societies where loneliness is more prevalent. On the other hand, given that loneliness seems generally uncommon in Norway, those who experience it might perceive themselves as outliers, prompting them to adopt ego defenses by embracing conspiracy theories. This may not be the case in societies where experiencing loneliness is more normative. Investigating the relationship between loneliness and conspiracist worldviews at the country level could bring further insights into the role of context, including cultural values (e.g., individualism-collectivism, relevant to both factors (1, 55, 59) and other nation-level variables such as general political climate, corruption, autocracy, or economic dysfunction (1, 50).

R2.5. References

Hettich, N., Beutel, M. E., Ernst, M., Schliessler, C., Kampling, H., Kruse, J., & Braehler, E. (2022). *Conspiracy endorsement and its associations with personality functioning, anxiety, loneliness, and sociodemographic characteristics during the COVID-19 pandemic in a representative sample of the German population*. *PLOS ONE*, 17(1), e0263301.

<https://doi.org/10.1371/journal.pone.0263301>

Poon, K., Chen, Z., & Wong, W. (2020). *Beliefs in conspiracy theories following ostracism*. *Personality and Social Psychology Bulletin*, 46(8), 1234-1246.

<https://doi.org/10.1177/0146167219898944>

Response to R2.5. We thank the Reviewer for indicating these references. We agreed that they are highly relevant and are now citing them in our manuscript.

Reviewer #3:

R3.1. Summary: *This article presents a longitudinal study that examines how loneliness across adolescence and adulthood predicts conspiracy beliefs in midlife. The authors use data from a population-based sample of Norwegians followed over 28 years and find that higher levels of*

loneliness in adolescence and its persistence over the life course are positively associated with conspiracy beliefs in midlife. The authors suggest that loneliness may lead to conspiracy beliefs as a way of coping with social isolation, seeking social connection, and protecting one's self-image. They also propose that interventions targeting loneliness might reduce or prevent conspiracy beliefs and their negative consequences for individuals and society.

The article addresses an important and timely topic of conspiracy beliefs and their psychological antecedents. The authors use a unique data set that allows for examining the long-term effects of loneliness on conspiracy beliefs over almost three decades. They employ rigorous statistical methods to test for measurement invariance, latent growth curve models, and robustness checks.

Response to R3.1. We thank the Reviewer for their positive evaluation. Please note that we have now improved model specification in line with most recent guidelines for second-order latent growth curves (please refer to **Response to R1.4** for details).

R3.2. *Nonetheless, the article has some limitations that could be addressed or acknowledged by the authors. First, the article does not account for potential confounding or moderating factors that could affect the relationship between loneliness and conspiracy beliefs (e.g., personality traits, cognitive abilities, social support, media exposure, or cultural values). The authors could include some of these variables as covariates or moderators in their models or discuss their possible role in future research.*

Response to R3.2. We thank the Reviewer for this comment, and we fully agree. In the presented analyses, we controlled for the relevant variables that were available in our dataset: age, sex, parental education, political orientation, depression, and anxiety symptoms. At the same time, we appreciate the Reviewer's comment regarding additional potential covariates (unfortunately not available in our dataset), which led us to extend various parts of our discussion.

First, we have now acknowledged and discussed the potential role of the variables listed by the Reviewer (p. 7):

As in any observational study, we cannot exclude the possibility that our results are confounded by third variables. We attenuated these risks by controlling for several potential confounds, namely age, sex, parental education, political orientation, depression and anxiety. Yet, other factors could play a role. For instance, individual differences such as personality traits, paranoid tendencies, and lower cognitive abilities, or experiences such as economic deprivation and media exposure might predispose individuals to both loneliness (48) and conspiracist worldviews (1, 49-51). However, including all relevant variables in one observational study is infeasible, and only rigorous experimental research can further eliminate possible spuriousness.

Second, as suggested, we now acknowledge that contextual and cultural factors may influence the effects (p. 8):

Investigating the relationship between loneliness and conspiracist worldviews at the country level could bring further insights into the role of context, including cultural values (e.g., individualism-collectivism, relevant to both factors (1, 55, 59) and other nation-level variables such as general political climate, corruption, autocracy, or economic dysfunction (1, 50).

R3.3. *Second, the article does not provide a clear theoretical framework or mechanism for explaining how loneliness leads to conspiracy beliefs. The authors could elaborate on their hypotheses and draw on existing theories or models of loneliness, social cognition, or motivated reasoning to support their arguments.*

Response to R3.3. We thank the Reviewer for this crucial comment. Indeed, the previous version of this paper presented the theoretical arguments in the discussion. We have now synthesized and moved these arguments, which indeed are in line with frameworks of motivated reasoning, to the introduction (pp. 2-3):

There are at least three theoretical reasons for this association. First, conspiracy beliefs may help make sense of one's loneliness (20) in a way that protects the ego, following general processes of motivated reasoning (21, 22). Sense-making and ego defense seem to be among the main psychological functions of conspiracy beliefs as per current theorizing (23) and could be particularly relevant for lonely people who generally seem hypervigilant to social threats and may use blame to deal with their own negative emotions (24). Conspiracy beliefs may preserve a positive self-image by shifting the blame for one's loneliness to malicious others (e.g., "I am not a failure but a victim of a conspiracy") (17). These beliefs may even enhance people's self-image by explaining their loneliness with their uniqueness (e.g., "I am alone because I understand things others do not understand") (17, 25). Second, lonely people may lack the social feedback that could correct their developing conspiracist views, and once these views are formed, such people may purposefully seek reinforcing feedback from other like-minded conspiracy believers (26, 27).

Lastly, loneliness may motivate people to adopt conspiracy beliefs in an attempt to gain community and a sense of social identity (1, 28). Several theoretical models describe loneliness as a motivational force across development (29-31). People who see themselves as lonely often (although not always; (32)) experience a powerful motivation to reconnect (29, 31), and seeking conspiracist communities might offer this opportunity. Online conspiracist groups in particular are easy to join, highly reinforcing and engaging, which may make them an accessible and suitable source of social nourishment and identity for socially isolated individuals (1, 27, 33). Indeed, individuals high in conspiracy beliefs are those who feel most socially isolated after unplugging from the internet (34). In this sense, conspiracy beliefs not only allow lonely

individuals to feel better about themselves but also provide them with communities that promote a sense of belonging.

R3.4. *Third, the article does not address the generalizability or replicability of their findings to other populations or contexts. The authors could discuss the limitations of their sample (e.g., predominantly ethnic Norwegians, relatively low levels of loneliness) and the specificity of their context (e.g., Norway as a highly functioning welfare society) and suggest how their results might differ or apply to other groups or settings.*

Response to R3.4. We agree with the Reviewer that the cultural context is important and deserves thorough discussion. As mentioned in our *Response to R3.2*, we have now delved deeper into matters of generalizability and replicability. Among other factors, we emphasize the relatively low national levels of loneliness and their statistical ramifications, as well as the societal consequences for those not aligning with this norm. We also discuss the significance of social welfare and propose how results might vary in different societies (p. 8):

The associations observed in this study might also depend on contextual factors. Here, it is important to note that our data were collected in Norway – a technologically advanced society with high levels of institutional trust (57) – and future research is needed to test the generalizability of our findings in other contexts (55). Norway can generally be described as a highly functioning welfare society with relatively low levels of loneliness (58). On the one hand, these low levels may mask, to some extent, the true size of associations between loneliness and conspiracist worldviews (i.e., due to floor effects). Thus, findings may be even more pronounced in societies where loneliness is more prevalent. On the other hand, given that loneliness seems generally uncommon in Norway, those who experience it might perceive themselves as outliers, prompting them to adopt ego defenses by embracing conspiracy theories. This may not be the case in societies where experiencing loneliness is more normative. Investigating the relationship between loneliness and conspiracist worldviews at the country level could bring further insights into the role of context, including cultural values (e.g., individualism-collectivism, relevant to both factors (1, 55, 59) and other nation-level variables such as general political climate, corruption, autocracy, or economic dysfunction (1, 50).

R3.5. *In sum, the article makes an interesting contribution to the literature on conspiracy beliefs and their psychological antecedents by using a longitudinal design and a large population-based sample. However, there are some methodological and conceptual limitations that weaken the study's validity and impact.*

Response to R3.5. We thank the Reviewer for the overall positive evaluation and the constructive feedback that helped strengthen our manuscript.

Reviewer #4:

R4.1. *The manuscript summarizes an interesting study on the association between loneliness (measured longitudinally throughout adolescence) and conspiracist worldviews later in life. The authors find that high levels of loneliness, especially if they persist over time, predict conspiracist worldviews.*

The study is interesting and novel and the methods used to address the research questions are advanced. Overall, I think this could be an interesting contribution to the existing literature on loneliness (and conspiracy beliefs).

Response to R4.1. We thank the Reviewer for their positive evaluation, as well as their thorough and constructive feedback.

R4.2. *However, I have some concerns about the manuscript in its current version.*

My biggest concern is related to the way how the variables were measured, particularly the fact that while loneliness was measured longitudinally over time, conspiracist worldview was not. In my opinion this is a major shortcoming of the present study because it hinders the authors to assess how longitudinal within-subject changes in one variable affect within-subject changes in the other, which would allow them to actually draw solid conclusions about predictions. As it is now, we don't know if individuals who showed conspiracist worldviews might have already held those at the beginning of data collection, which might have been the reason why they became lonely in the first place. Thus, there is no way to test if the causality is actually in the other direction. However, I don't think there is a way how the authors can address this alternative explanation given the data. I therefore suggest that the authors thoroughly rephrase the manuscript to tone down the implied causality of the results. At the moment the whole manuscript, and especially the discussion, heavily imply causality in the effects, which I don't think is warranted given the data.

Response to R4.2. We thank the Reviewer for this comment. We agree that the fact that conspiracy mentality is only available at T5 is a limitation of this study, making it impossible to test the reverse time precedence or feedback loops. We also agree that, since the study is observational, our results do not warrant causal claims. Accordingly, we have taken various steps to address the Reviewer's valid comment. First, we have removed any causal claims in the paper. Second, we now discuss more extensively the limitations of our analyses, including the role of potential confounding variables (p. 7):

As in any observational study, we cannot exclude the possibility that our results are confounded by third variables. We attenuated these risks by controlling for several potential confounds, namely age, sex, parental education, political orientation, depression and anxiety. Yet, other factors could play a role. For instance, individual differences such as personality traits, paranoid tendencies, and lower cognitive abilities, or experiences such as economic deprivation and media exposure might predispose individuals to both loneliness (48) and conspiracist worldviews (1, 49-51). However, including all relevant variables in one observational study is infeasible, and only rigorous experimental research can further eliminate possible spuriousness.

Third, when discussing the potential interventions, we now remind readers of the need to replicate our findings within different contexts and with different designs and generally formulate our suggestions more cautiously (p. 9):

Although we encourage the replication of our findings in other contexts and with different designs to strengthen causal inferences, this study suggests that interventions targeting loneliness could be useful to reduce conspiracy beliefs and their societal repercussions.

Additionally, we would like to note that despite the limitations rightfully pointed out by the Reviewer, we believe our test goes beyond the level of cross-sectional correlation. We now discuss this point in the manuscript (pp. 6 – 7):

Methodologically, our study has one key limitation: as conspiracist worldviews represent a relatively recent construct in psychological research, controlling for participants' initial levels in 1992 was impossible. Even so, we believe that our finding of a significant association between loneliness in adolescence and conspiracy mentality in midlife is a novel result that goes far beyond earlier cross-sectional findings. Showing that the estimated levels of loneliness in adolescence are predictive of conspiracy mentality in midlife, almost three decades later and accounting for later developments of loneliness, suggests that adolescent loneliness is related to conspiracy mentality over substantially long timeframes. Notably, this finding held even when controlling, in adolescence, for several potential confounders closely related to conspiracy mentality, such as political orientation (5) or anxiety and depression (1). Combined with the significant association of the slope of loneliness to conspiracy mentality in midlife, this finding indicates that loneliness and conspiracy mentality are systematically related to each other from adolescence to midlife in ways consistent with the theoretical notion that conspiracist worldviews reflect sense making and ego defenses adopted in response to loneliness (19, 20, 22). The unavailability of earlier measurements of conspiracy worldviews is a limitation nevertheless. For instance, it prevents us from testing a plausible feedback loop: that adopting a conspiracist worldview might further exacerbate loneliness. Indeed, conspiracists might be excluded from many social groups or ostracized, which would indicate that turning to conspiracy beliefs is not an adaptive strategy to cope with loneliness.

R4.3. *Some parts of the data analysis were not clear to me, and I think it would be important to add more information there:*

Response to R4.3. We very much appreciate that the Reviewer took the time to thoroughly read our analyses and point out aspects that were less clear. Please note that we have now improved the specification of our statistical models in line with most recent guidelines (Wickrama, Lee, O’Neal, & Lorenz, 2022; please refer to **Response to R1.4** for details). By consequence, the shape of the estimated average trajectory of loneliness changed somewhat, and we have revised the text and figures in the manuscript accordingly.

R4.4. *-lines 81-82 in the results section: the authors describe how they regressed loneliness on depression and anxiety scores thereby removing variance in loneliness explained by mental health problems. I don’t fully understand how and why this regression removes variance coming from mental health problems and I would imagine other readers might find this hard to understand as well. It would be good to provide more explanation on this approach. Given that depression/anxiety are highly correlated with loneliness at each time point (as seen in table S2), this is not a trivial issue so it would be great to see more information on this part of the analysis. Related to that, I was wondering if there are interactions between loneliness and depression/anxiety that might explain developments of conspiracist world views. As in, there might be a difference in conspiracist worldviews between people who are lonely and show high scores of depression/anxiety compared to those who are lonely but have low depression/anxiety. It would be interesting to explore such interactions.*

Response to R4.4. We fully agree that the previous description of the procedure was unclear. In the revision, we have now clarified this part (p. 5):

Separate lines of research have linked conspiracy beliefs (1, 26) and loneliness (40) to psychopathology. To rule out that the associations with loneliness were artifacts of underlying psychopathology, we included symptoms of depression and anxiety measured by a short version of the Hopkins Symptom Checklist (41) (.83 \leq $a_{SCL} \leq$.91) as time-varying covariates in the model. Specifically, when estimating loneliness growth curves, we regressed loneliness within each timepoint on depression and anxiety scores (42); $\chi^2(324) = 2,134.06, p < .001$; CFI = .907, SRMR = .038; RMSEA = .050, 95% CI [.048, .052], $p_{close} = .424$. In this model, the intercept represents the baseline and the slope represents the growth in loneliness that is not attributable to temporal changes in symptoms of depression and anxiety. Even after controlling for these symptoms, the initial level ($b_{intercept} = .11, 95\% \text{ CI } [.06, .16], p = .001$) and the trajectory ($b_{slope} = .11, 95\% \text{ CI } [.03, .18], p = .015$) of loneliness remained positively related to conspiracy worldviews.

We would like to clarify that regressing the focal time-varying predictor (in our case, loneliness) on a time-varying covariate (TVC; in our case, HSCL score) within each wave is a standard procedure in latent growth curve modeling, sometimes referred to in the literature as the TVC

model (Grimm, 2007). This approach is used to control for potentially relevant variables/confounders that vary over time (secondary process) together with the focal predictor (primary process). The inclusion of TVCs is thought to better approximate the true growth process, on the condition that the TVC is related to the focal predictor (Mund, Johnson & Nestler, 2021). Thus, the correlation between the TVC (HSCL scores) and the focal predictor (conspiracy mentality) is a pre-requisite for this kind of analysis.

For a detailed explanation how this procedure is implemented, we cite Grimm (2007, p. 332) below, now also referenced in the main manuscript to provide interested readers with further information (p. 5):

The latent growth curve with a TVC model has become a popular model for investigating multiple developmental processes. In the bivariate TVC model one of the developmental processes is considered primary as change in this variable is modeled, whereas the secondary process is seen as a control variable. For the TVC model, we again assume we have repeatedly measured two variables, A and D, for a sample of $n = 1$ to N participants. The TVC model with A as the primary and D as the secondary process can be written as

$$A[t]_n = 1 \cdot g_{0n} + B[t] \cdot g_{1n} + \beta \cdot D[t]_n + 1 \cdot s[t]_n,$$

where $A[t]_n$ are the repeated measurements of the observed variable A, g_{0n} is a latent intercept, g_{1n} is a latent slope, $B[t]$ is a vector of basis coefficients, $D[t]_n$ are the repeated measurements of the observed variable D (the time-varying covariate), β is the effect of A regressed on D at time t , and $s[t]_n$ is a time-specific residual. (...) The TVC model enables the researcher to examine the growth in the primary variable while controlling for the TVC. The research questions that can be answered by the TVC model for depression and achievement [*the example used in the cited paper*], assuming that achievement is the primary and depression is the secondary process, are: (1) How are the time-varying scores of depression related to the time-varying scores of achievement while controlling for interindividual differences in intraindividual change in achievement? [...] (3) How does achievement change over time when controlling for depression over time?

When it comes to interactions, we think this is an interesting idea. Unfortunately, our data are not best suited to test it. In latent growth curve modeling, interactions between TVCs are tested differently than in autoregressive models and are generally interpreted in terms of interacting trajectories and intercepts (e.g., the pace of increase/decrease of a trajectory is predicted by the intercept of a different construct). Provided that, as the Reviewer points out, loneliness and HSCL scores are strongly related, such an analysis would only show that both trajectories are correlated, and the intercept of one trajectory predicts the slopes of the other, and vice versa. This would not be informative for the proposed hypotheses. Testing them would require a completely different methodological approach that, with our data structure, would almost entirely remove the longitudinal aspect (e.g., testing if interaction terms at T1 predict conspiracy worldviews at T5, and omitting the growth processes). Such a test would not optimally utilize the

unique temporal dimension of our data, therefore we refrained from conducting it. However, we thank the Reviewer for highlighting this possibility.

References:

Mund, M., Johnson, M. D., & Nestler, S. (2021). Changes in size and interpretation of parameter estimates in within-person models in the presence of time-invariant and time-varying covariates. *Frontiers in psychology, 12*, 666928.

Grimm, K. J. (2007). Multivariate longitudinal methods for studying developmental relationships between depression and academic achievement. *International Journal of Behavioral Development, 31*(4), 328-339.

R4.5. *-I don't understand why conspiracy worldview was regressed on the linear slope of loneliness but not the quadratic slope. The authors say that only the linear slope was used due to its high correlation with the quadratic slope to avoid multicollinearity, which I understand, but why did the authors not choose the quadratic slope instead of the linear slope in that case given that the quadratic slope model fitted the data best?*

Response to R4.5. We thank the Reviewer for drawing our attention to this aspect. It is important to clarify that the two slopes do not refer to alternative trajectories (linear vs. quadratic) but instead describe the characteristics of a single trajectory. The linear slope is the 1st degree polynomial that indicates whether the trajectory is decreasing or increasing; the quadratic slope is the 2nd degree polynomial that indicates the acceleration/deceleration of this decrease/increase over the subsequent timepoints. Both pieces of information are applied, within the same model, to the same line that can be expressed as:

trajectory = intercept (where does the line start?) + linear slope (does it increase or decrease?) + quadratic slope (is this increase/decrease similar between the different timepoints?) + error

We hope this clarifies that by regressing the outcome on the linear slope only, we did not choose a linear trajectory over a curvilinear trajectory. Instead, we explain the variation in the outcome (conspiracy theory) with the extent to which the trajectory of loneliness decreases for our participants, instead of the extent to which the pace of this decrease changes.

Lastly, we would like to note that Reviewer 1 raised a somewhat similar point, thus an even more detailed explanation of the strong negative correlation between both slopes can be found in our **Response to R1.5 and R.6.**

R4.5. *-it would be interesting to see data on how many people in the sample actually held conspiracy beliefs. I couldn't find this information in the manuscript or supplement (but might have missed it).*

Response to R4.5. We agree with the Reviewer that this is an interesting question. Unfortunately, our questionnaire did not include any items directly addressing the belief in specific conspiracy theories. Based on the mean ($M = 3.78$) and standard deviation ($SD = 1.31$) of the general conspiracy mentality measure reported in Supplementary Materials (Table S2), 16% of the participants scored above 5 (i.e., higher than 1 SD above the mean) on the 7-point scale. We may thus assume that participants who exceeded this cutoff are likely to interpret events in a conspiracist manner, but we lack the data to test this.

R4.6. Introduction/Discussion:

-I was missing more background/ a theoretical framework on conspiracy worldviews in the introduction.

Response to R4.6. We thank the Reviewer for this comment. Indeed, the previous version of this paper only included theoretical arguments in the discussion. We have now moved a streamlined presentation of these arguments to the introduction (pp. 2 – 3):

There are at least three theoretical reasons for this association. First, conspiracy beliefs may help make sense of one's loneliness (19) in a way that protects the ego, following general processes of motivated reasoning (20, 21). Sense-making and ego defense seem to be among the main psychological functions of conspiracy beliefs (22) and could be particularly relevant for lonely people who generally seem hypervigilant to social threats and may use blame to deal with their own negative emotions (23). Conspiracy beliefs may preserve a positive self-image by shifting the blame for one's loneliness to malicious others (e.g., "I am not a failure but a victim of a conspiracy") (17). These beliefs may even enhance people's self-image by explaining their loneliness with their uniqueness (e.g., "I am alone because I understand things others do not understand") (17, 24). Second, lonely people may lack the social feedback that could correct their developing conspiracist views, and once these views are formed, such people may purposefully seek reinforcing feedback from other like-minded conspiracy believers (25, 26).

Lastly, loneliness may motivate people to adopt conspiracy beliefs in an attempt to gain community and a sense of social identity (1, 27). Several theoretical models describe loneliness as a motivational force across development (28-30). People who see themselves as lonely often (although not always; (31)) experience a powerful motivation to reconnect (28, 30), and seeking conspiracist communities might offer this opportunity. Online conspiracist groups in particular are easy to join, highly reinforcing and engaging, which may make them an accessible and suitable source of social nourishment and identity for socially isolated individuals (1, 26, 32). Indeed, individuals high in conspiracy beliefs are those who feel most socially isolated after

unplugging from the internet (33). In this sense, conspiracy beliefs not only allow lonely individuals to feel better about themselves but also provide them with communities that promote a sense of belonging.

R4.7. *For example, do we assume that people who hold conspiracy beliefs do it for similar reasons even if the specific conspiracies differ in terms of content?*

Response to R4.7. This is an important question. As raised above, we measured conspiracy mentality in the current paper; not belief in specific conspiracy theories. In the revision, however, we now discuss how the focus on specific conspiracy theories may be meaningful (p. 8):

Further, research might also explore how effects differ when examining beliefs in specific conspiracy theories, oftentimes deeply rooted in culture and sociopolitical divides, as opposed to the broad conspiracist worldview our study focused on. Our broad approach avoids catering exclusively to specific cultural contexts or groups, thus enhancing the generalizability of our findings. However, it is possible that loneliness promotes some conspiracy beliefs more than others. For instance, conspiracy theories rooted in a nation's shared historical trauma are likely more commonly endorsed by its members (49, 58) and, thus, might be more appealing to those seeking social connection.

R4.8. *-the authors state in the discussion (line 101) that individuals who are lonely experience a powerful motivation to reconnect. However, many studies within the loneliness literature show that lonely individuals often show lower social approach motivation. It would be nice to have a more nuanced discussion on why loneliness might affect conspiracy worldviews.*

Response to R4.8. We thank the Reviewer for pointing us to these opposing findings. We now acknowledge this inconsistency of results in previous work (p. 3):

Several theoretical models describe loneliness as a motivational force across development (29-31). People who see themselves as lonely often (although not always; 32) experience a powerful motivation to reconnect (29, 31), and seeking conspiracist communities might offer this opportunity.

REVIEWER COMMENTS

Reviewer #1 (Remarks to the Author):

My comments have been addressed by the authors. There is one issue where I continue to disagree slightly, though. In their response, the authors write:

"our study only fulfills two conditions of causality - association and time precedence - but it cannot fulfill the third condition, nonspuriousness"

I slightly disagree with the time precedence bit - we just don't know whether loneliness precedes CM, as we have no prior measures of CM. This is obvious for the correlation of intercepts, but applies also to the slope prediction. It is conceivable that CM has similar slopes but logged on wave back, making CM the actual precursor or loneliness trajectories. This may or may not seem likely, my nit-picky point is: that we do not know.

Reviewer #2 (Remarks to the Author):

I would like to thank you for your carefully considered responses to my review comments. Despite some of them potentially having personal views behind them I think you did well to bring them up as alternative/complementary perspectives.

Reviewer #3 (Remarks to the Author):

Based on my review of the attached rebuttal letter and reviewer comments, it seems the authors have done a good job addressing the concerns raised, especially those I raised in my previous review.

Specifically, the authors have now acknowledged this limitation and discussed the potential role of factors like personality, cognitive abilities, economic deprivation, and media exposure. They also noted the challenges of including all relevant variables in one study, and suggested experimental research could help rule out confounds.

The authors have added several paragraphs to the introduction summarizing mechanisms like motivated reasoning, sense-making, ego defense, lack of social feedback, and seeking community/identity. This helps explain their hypotheses better.

Overall, the authors appear to have thoroughly addressed my previous concerns about confounding factors, theoretical mechanisms, and generalizability. They acknowledged limitations, suggested future research directions, and cited relevant literature. The changes seem to strengthen the validity of the study and contextualize the findings appropriately. I would agree the revised manuscript is much improved.

Reviewer #4 (Remarks to the Author):

Bierwaczonok et al. "Loneliness trajectories over three decades predict conspiracist worldviews in midlife"

The authors have done a good job in addressing my concerns. Particularly the added information on the methods is very helpful.

However, I don't think my main concern (i.e., the fact that conspiracist world views was only measured at the end) was addressed adequately in the current version of the manuscript.

While the authors did include this issue as a limitation in the discussion section, the language in the

rest of the manuscript still heavily implies causality of effects (i.e., loneliness causing conspiracist world views). For example, in the discussion, the authors are still suggesting that their findings imply that loneliness interventions could reduce conspiracist world views.

This is problematic because even if the authors find a positive prediction of conspiracist world views from loneliness in adolescence (and slope of loneliness), we simply do not know the true relationship of the variables given the lack of measurements of conspiracist world views at earlier time points. It might as well be that the variables are highly correlated at each time point throughout development (perhaps even with the opposite causal relationship, i.e., conspiracist world views causing loneliness). By only looking at parts of the data, such as the association of one variable (loneliness) at an earlier time point and the other (conspiracist world views) at a later time point, one would expect to find a positive prediction of one variable from the other, even if the true relationship between the variables was different.

Because there is no way of testing any other association between the variables other than the one the authors tested for, we simply cannot draw any strong conclusions about their relationship.

This, in my view, represents a major limitation of how we can interpret the results of this analysis, but I do not think the authors clarify enough the severity of this limitation. The language in the introduction and discussion is still heavily causal.

Also, the authors currently argue that they control for confounders „Notably, this finding held even when controlling for several potential confounders closely related to conspiracy mentality, such as political orientation (5) and anxiety and depression (1).“

However, political orientation and (especially) anxiety and depression are related to a variety of other constructs, so controlling for these hardly controls for the possibility that previously held conspiracist world views (during adolescence) are driving the results.

I think the authors need to substantially rephrase the manuscript to account for this issue.

Alternatively, if the authors disagree, they should present a convincing argument for why their current interpretation is valid.

Minor:

The authors now added a statement that loneliness does not always lead to an increased motivation to reconnect as a side note. However, there is actually more evidence showing that loneliness is associated with social withdrawal than the other way around (see studies from the Cacioppo group) so I think this statement is still misleading (or at least not correctly representing the current state of the literature).

Reviewer #1:

***R1.** My comments have been addressed by the authors. There is one issue where I continue to disagree slightly, though. In their response, the authors write:*

"our study only fulfills two conditions of causality - association and time precedence - but it cannot fulfill the third condition, nonspuriousness"

I slightly disagree with the time precedence bit - we just don't know whether loneliness precedes CM, as we have no prior measures of CM. This is obvious for the correlation of intercepts, but applies also to the slope prediction. It is conceivable that CM has similar slopes but logged on wave back, making CM the actual precursor or loneliness trajectories. This may or may not seem likely, my nit-picky point is: that we do not know.

***Response to R1.** We thank the Reviewer for their supportive and constructive feedback through the review process. The comment about temporal precedence relates to the "response to reviewers" letter from the last round rather than to the manuscript itself. However, the point dovetails with questions about causality raised also by Reviewer 4. A full engagement with this issue can be found in our **Response to R4.2**.*

Reviewer #2:

***R2.** I would like to thank you for your carefully considered responses to my review comments. Despite some of them potentially having personal views behind them I think you did well to bring them up as alternative/complementary perspectives.*

***Response to R2.** We very much appreciate the Reviewer's positive feedback.*

Reviewer #3:

***R3.** Based on my review of the attached rebuttal letter and reviewer comments, it seems the authors have done a good job addressing the concerns raised, especially those I raised in my previous review.*

Specifically, the authors have now acknowledged this limitation and discussed the potential role of factors like personality, cognitive abilities, economic deprivation, and media exposure. They

also noted the challenges of including all relevant variables in one study, and suggested experimental research could help rule out confounds.

The authors have added several paragraphs to the introduction summarizing mechanisms like motivated reasoning, sense-making, ego defense, lack of social feedback, and seeking community/identity. This helps explain their hypotheses better.

Overall, the authors appear to have thoroughly addressed my previous concerns about confounding factors, theoretical mechanisms, and generalizability. They acknowledged limitations, suggested future research directions, and cited relevant literature. The changes seem to strengthen the validity of the study and contextualize the findings appropriately. I would agree the revised manuscript is much improved.

Response to R3. We thank the Reviewer for their positive evaluation.

Reviewer #4:

R4.1. *The authors have done a good job in addressing my concerns. Particularly the added information on the methods is very helpful.*

Response to R4.1. We thank the Reviewer for their positive evaluation.

R4.2. *However, I don't think my main concern (i.e., the fact that conspiracist world views was only measured at the end) was addressed adequately in the current version of the manuscript. While the authors did include this issue as a limitation in the discussion section, the language in the rest of the manuscript still heavily implies causality of effects (i.e., loneliness causing conspiracist world views). For example, in the discussion, the authors are still suggesting that their findings imply that loneliness interventions could reduce conspiracist world views. This is problematic because even if the authors find a positive prediction of conspiracist world views from loneliness in adolescence (and slope of loneliness), we simply do not know the true relationship of the variables given the lack of measurements of conspiracist world views at earlier time points. It might as well be that the variables are highly correlated at each time point throughout development (perhaps even with the opposite causal relationship, i.e., conspiracist world views causing loneliness).*

By only looking at parts of the data, such as the association of one variable (loneliness) at an earlier time point and the other (conspiracist world views) at a later time point, one would expect to find a positive prediction of one variable from the other, even if the true relationship between the variables was different.

Because there is no way of testing any other association between the variables other than the one the authors tested for, we simply cannot draw any strong conclusions about their relationship. This, in my view, represents a major limitation of how we can interpret the results of this analysis, but I do not think the authors clarify enough the severity of this limitation. The language in the introduction and discussion is still heavily causal.

Also, the authors currently argue that they control for confounders „Notably, this finding held even when controlling for several potential confounders closely related to conspiracy mentality, such as political orientation (5) and anxiety and depression (1).“

However, political orientation and (especially) anxiety and depression are related to a variety of other constructs, so controlling for these hardly controls for the possibility that previously held conspiracist world views (during adolescence) are driving the results..

Response to R4.2. We thank the Reviewer for pressing us on this issue. We have now proofread the entire manuscript once more to ensure that any terms with a causal connotation (e.g., “predict”) were removed from the abstract, the results, and the discussion. In the introduction, we only found causal language in passages describing the theoretical mechanisms linking loneliness and conspiracy beliefs. We refrained from editing these passages because the underlying theory does indeed present a causal case, even if our dataset itself cannot establish causality. Consistent with this, our hypotheses speak of a “positive association” between loneliness trajectories and conspiracist worldviews rather than a causal effect.

Additionally, we have restructured and edited the discussion to further emphasize that interventions targeting loneliness will only be viable once the causal link between loneliness and conspiracy beliefs is established experimentally. Specifically, we extended and moved the discussion of the implications of omitting earlier measurements of conspiracy beliefs closer to the passage about interventions, we connected these paragraphs, and we removed or rephrased unjustifiably strong statements about the implications of the data for interventions. We also removed the sentence regarding confounders. Finally, we now end the manuscript by reiterating the need for testing loneliness interventions experimentally. As a result, these paragraphs read as follows (pp. 8-9):

Methodologically, our study has one key limitation: as conspiracist worldviews represent a relatively recent construct in psychological research, controlling for participants’ initial levels in 1992 was impossible. Even so, we believe that our finding of a significant association between loneliness in adolescence and conspiracy mentality in midlife is a novel result that goes beyond earlier cross-sectional findings (16, 35). Showing that the estimated levels of loneliness in adolescence are associated with conspiracy mentality in midlife, almost three decades later and accounting for later developments of loneliness, suggests that adolescent loneliness is related to conspiracy mentality over substantially long timeframes. Combined with the significant association of the slope of loneliness to conspiracy mentality in midlife, this finding may suggest that loneliness and conspiracy mentality are systematically related to each other from adolescence to midlife in ways

consistent with the theoretical notion that conspiracist worldviews reflect sense making and ego defenses adopted in response to loneliness (19, 20, 22). The unavailability of earlier measurements of conspiracy worldviews is a major limitation, nevertheless. Namely, this study prevents us from testing a reverse causal direction: that adopting a conspiracist worldview might further exacerbate loneliness. Indeed, people who express conspiracy theories early in life might be excluded from social groups (1) which could lead to feelings of loneliness.

This considered, future research would be well advised to attempt a replication of our results in other contexts, and with experimental designs allowing for causal inferences. If successful, such replications would suggest that interventions targeting loneliness could be useful to reduce conspiracy beliefs and their societal repercussions. So far, the results of psychological interventions focusing primarily on cognitive processes (e.g., pre-bunking, debunking, cognitive inoculation) have been insufficient on their own to fully counter conspiracist worldviews (1, 12, 60). There might, however, exist an alternative, complementary pathway to prevent conspiracy beliefs, one that leads via socio-affective processes. On one hand, previous research showed that the link between experimentally manipulated social exclusion (i.e., ostracism) and conspiracist thinking can be mitigated (51), and the same may be the case for loneliness. On the other hand, targeting loneliness and fostering social connection is known to be effective: it helps prevent other adverse outcomes, including somatic and mental health problems or even mortality risks, for a myriad of different social groups (61-64). Therefore, instead of concentrating solely on cognitive factors, research may test experimentally whether reducing people's loneliness is a way to counter the onset of conspiracist worldviews and their societal repercussions.

R4.3. *The authors now added a statement that loneliness does not always lead to an increased motivation to reconnect as a side note. However, there is actually more evidence showing that loneliness is associated with social withdrawal than the other way around (see studies from the Cacioppo group) so I think this statement is still misleading (or at least not correctly representing the current state of the literature).*

Response to R4.3. We appreciate this comment. We have now rephrased this statement as follows, and we added one more reference to a study from the Cacioppo lab:

Some people who see themselves as lonely may experience a motivation to reconnect (28, 30), and seeking conspiracist communities might offer this opportunity. Online conspiracist groups in particular are easy to join, highly reinforcing and engaging, which may make them an accessible and suitable source of social nourishment and identity for socially isolated individuals (1, 26, 31). Indeed, individuals high in conspiracy beliefs are those who feel most socially isolated after unplugging from the internet (32). However, it should be noted that other lines of research suggest that loneliness is associated with social withdrawal rather than a motivation to reconnect (33, 34), which would make this a less plausible mechanism underpinning the link between loneliness and conspiracy beliefs than the previously described mechanisms of ego protection and lack of corrective feedback.

REVIEWERS' COMMENTS

Reviewer #1 (Remarks to the Author):

I agree with the qualifications added to the discussion and have no other objections.

Reviewer #4 (Remarks to the Author):

The authors have done a great job in revising the manuscript to refrain from causal language. One minor suggestion I have for the introduction is to add that conspiracist worldviews can lead to loneliness too (or that there might be a bi-directional relationship), when discussing cross-sectional findings on the association between loneliness and conspiracist world views. Perhaps around lines 69-70 where the authors state that longitudinal research is scarce, as it would also highlight why longitudinal evidence is needed in addition to cross-sectional evidence.

Otherwise I have no further suggestions and recommend that the paper is published.

Reviewer #4:

R4.1. *The authors have done a great job in revising the manuscript to refrain from causal language.*

Response to R4.1. We are glad that the Reviewer found our revisions satisfactory.

R4.2. *One minor suggestion I have for the introduction is to add that conspiracist worldviews can lead to loneliness too (or that there might be a bi-directional relationship), when discussing cross-sectional findings on the association between loneliness and conspiracist world views. Perhaps around lines 69-70 where the authors state that longitudinal research is scarce, as it would also highlight why longitudinal evidence is needed in addition to cross-sectional evidence.*

Response to R4.2. We thank the Reviewer for this suggestion. The extent to which our data speak to causal effects in either direction had been subject to an extensive discussion throughout the review process, and with this in mind, we would rather refrain from suggesting that our results can clarify whether loneliness leads to conspiracy worldview or the opposite. However, we agreed with the Reviewer that the need for longitudinal research should have been articulated better, and we have now edited the lines indicated by the Reviewer as follows:

So far, however, research on the link between loneliness and conspiracy beliefs ¹⁶, and in particular longitudinal research that could clarify how this link plays out over time, is scarce ¹.

Response to R4.3. *Otherwise I have no further suggestions and recommend that the paper is published.*

Response to R4.3. We thank the Reviewer for their positive recommendation.